# A 0.2 T–0.4 T Static Magnetic Field Improves the Bone Quality of Mice Subjected to Hindlimb Unloading and Reloading Through the Dual Regulation of BMSCs via Iron Metabolism

**DOI:** 10.3390/ijms252313136

**Published:** 2024-12-06

**Authors:** Jianping Wang, Chenxiao Zhen, Gejing Zhang, Zhouqi Yang, Peng Shang

**Affiliations:** 1School of Life Sciences, Northwestern Polytechnical University, Xi’an 710072, China; wangjianpingemail@mail.nwpu.edu.cn (J.W.); zhenchenxiao@mail.nwpu.edu.cn (C.Z.); zgj@mail.nwpu.edu.cn (G.Z.); yangzhouqi@nwpu.edu.cn (Z.Y.); 2Key Laboratory for Space Bioscience and Biotechnology, Institute of Special Environmental Biophysics, School of Life Sciences, Northwestern Polytechnical University, Xi’an 710072, China; 3Research & Development Institute, Northwestern Polytechnical University, Shenzhen 518057, China

**Keywords:** 0.2 T–0.4 T SMF, HLU reloading, bone quality, iron metabolism, BMSCs

## Abstract

Osteoporosis is the most prevalent metabolic bone disease, especially when aggravated by aging and long-term bed rest of various causes and also when coupled with astronauts’ longer missions in space. Research on the use of static magnetic fields (SMFs) has been progressing as a noninvasive method for osteoporosis due to the complexity of the disease, the inconsistency of the effects of SMFs, and the ambiguity of the mechanism. This paper studied the effects of mice subjected to hindlimb unloading (UL, HLU) and reloading by the 0.2 T–0.4 T static magnetic field (MMF). Primary bone marrow mesenchymal stem cells (BMSCs) were extracted to explore the mechanism. Eight-week-old male C57BL/6 mice were used as an osteoporosis model by HLU for four weeks. The HLU recovery period (reloading, RL) was carried out on all FVEs and recovered in the geomagnetic field (45–64 μT, GMF) and MMF, respectively, for 12 h/d for another 4 weeks. The tibia and femur of mice were taken; also, the primary BMSCs were extracted. MMF promoted the recovery of mechanical properties after HLU, increased the number of osteoblasts, and decreased the number of adipocytes in the bone marrow. MMF decreased the total iron content and promoted the total calcium content in the tibia. In vitro experiments showed that MMF promoted the osteogenic differentiation of BMSCs and inhibited adipogenic differentiation, which is related to iron metabolism, the Wnt/β-catenin pathway, and the PPARγ pathway. MMF accelerated the improvement in bone metabolism and iron metabolism in RL mice to a certain extent, which improved the bone quality of mice. MMF mainly promoted osteogenic differentiation and reduced the adipogenic differentiation of BMSCs, which provides a reliable research direction and transformation basis for the osteoporosis of elderly, bedridden patients and astronauts.

## 1. Introduction

Osteoporosis is a kind of bone metabolic disease associated with iron overload, which is characterized by low bone mass and the destruction of bone microstructure, thus resulting in bone fragility [1]. Patients who are bedridden for many years cannot get enough exercise, which results in the degeneration of the skeletal muscle system and bone loss [2]. In addition to this kind of osteoporosis population, there is also concern about the bone loss of astronauts. Astronauts suffer from serious bone loss after space flights [3]. The composition and microstructure of bones are complex and adjust over time in response to changes in the internal and external environment. Some factors, such as aging and estrogen deficiency, affect these components, ultimately leading to decreased bone strength and fracture toughness [4]. The biomechanical properties of bone depend on the quantity and quality of the skeletal material and also on the spatial arrangement of materials in space [5]. Changes in bone mechanical properties can be explained by the functional adaptation of bone structure and age-related degradation, both of which are directly related to bone remodeling [6]. At present, the treatment of osteoporosis mainly includes drug therapy and physical therapy [7,8,9,10]. However, the complicated etiology and treatment of osteoporosis, as well as the side effects of drugs, make it extremely urgent to obtain more efficient measures to defeat it.

Electrical stimulation for bone disorders began in the 19th century. In the 1970s, Brighton inserted cathodes into fracture sites to promote healing through a constant microcurrent of 10 μA [11]. In 1974, Bassett reported that low-frequency pulsed electromagnetic fields could effectively promote fracture healing [12]. In 1977, Bassett was the first to use pulsed electromagnetic fields to treat 127 patients with delayed fracture union and nonunion. This non-operative treatment has resulted in satisfactory results for more than 70% of patients with delayed union and nonunion [13]. Scholars hold a positive attitude toward the clinical application of pulsed electromagnetic fields, but the mechanism of the electric effect, magnetic effect, or thermal effect contained has not been determined. Therefore, studies on the biological effects of the static magnetic field (SMF) as a single physical source on tissues and cells have emerged in recent years. The stability of the implant on the side of the NdFeB magnet with a strength of 50 mT was better than that on the non-magnetic side. After 3 continuous months of exposure, the clinical results showed a positive correlation between the magnetic field and bone healing [14].

Iron is one of the essential elements in the human body and participates in many physiological processes. Under normal physiological conditions, iron metabolism homeostasis is mainly maintained by systemic iron metabolism and cellular iron metabolism [15]. Systemic iron metabolism mainly occurs through the absorption and efflux of iron in the small intestine, blood transport, tissue utilization, liver storage and phagocytosis, and the circulation of macrophages to achieve homeostasis [16]. Cellular iron metabolism mainly occurs through the process of iron uptake, utilization, and excretion [17]. The links of the iron metabolism regulatory network complement each other and depend on each other. In recent years, more and more studies have proven that SMFs play an important role in the regulation of iron metabolism in organisms. In 1974, the influence of SMFs on the change in iron content in animal tissues and organs was first proposed [18]. The temporary exposure of a 128 mT SMF could reduce the amount of iron in the serum of Wistar rats, while the transferrin concentration significantly increased [19]. Moreover, a 16 T SMF promotes the differentiation of MC3T3-E1 cells and the expression of FPN, inhibiting the expression of DMT1 [20]. Iron chelation can relieve the symptoms of osteoporosis to a certain extent and has a positive effect on its treatment [21].

Mesenchymal stem cells are multipotent adult stem cells that exist in almost all postnatal organs and tissues, such as bone marrow, adipose tissue, liver, muscle, umbilical cord, etc. Bone marrow mesenchymal stem cells (BMSCs) were first discovered by Freidenstein in 1976 [22]. BMSCs can differentiate into osteoblasts and adipocytes and play an important role in maintaining bone homeostasis. The decrease in the number or biological activity of BMSCs causes an imbalance of the dynamic reconstruction of bone tissue and, finally, leads to osteoporosis/osteoporotic fracture [23]. When an osteoporotic fracture occurs, BMSCs can promote the generation of new bone from damaged bone tissue through various mechanisms, such as cell secretion, immune regulation, gene regulation, and the activation of the signaling pathway, and also improve the regeneration ability of local blood vessels, providing a good microenvironment for bone tissue reconstruction. It can be seen that there is a relationship between maintaining the normal biological activity of BMSCs and the treatment of osteoporosis [24,25]. The use of a 200 mT SMF significantly increased the proliferation rate of rBMSCs, ALP activity, and osteogenic protein expression, and the levels of matrix mineralization were significantly higher than those of the control group [26]. However, it has also been reported that with the increase in the SMF intensity (4, 7, 15 mT) and time, the survival rate and proliferation rate of BMSCs were reduced [27]. It has also been reported that a 0.2 T–0.6 T SMF inhibited the differentiation of MSCs into adipocytes [28]. It has also been shown that 0.5 T SMF treatment for 7 days slightly reduced the cell viability and proliferation ability of adipose-derived stem cells and inhibited adipogenic and osteogenic differentiation, but it did not affect DNA integrity [29]. The biological effects and mechanisms of BMSCs under SMF with different characteristics need to be further explored and systematically verified. Therefore, based on the refractory nature of osteoporosis, the relationship between iron overload and osteoporosis, and also the key role of BMSCs in bone development, we wanted to explore the biological effects of SMF on HLU reloading mice (in a simulated osteoporosis treatment period), and the role and mechanism of iron and iron metabolism. As an osteoblastic progenitor cell, how does BMSCs play a role in the treatment of osteoporosis by SMF and what is the mechanism? This basic research has important innovation and medical transformation significance for the application of SMF in the clinical treatment of osteoporosis.

## 2. Results

### 2.1. MMF Improved the Mechanical Properties of the Tibia of RL Mice and the Serum Level of the Bone Metabolism Molecular Components

Through the three-point bending test, the mechanical characteristics of the tibia in each group were measured. Upon normalizing the internal and external diameter data of the tibia cross-section, it was found that in contrast to the RL group, MMF enhanced the mechanical properties of the tibia. This encompassed aspects such as stiffness, ultimate stress, ultimate load, elastic modulus, yield stress, and overall toughness (Figure 1C), with the stiffness index exhibiting a significant variation. When compared to the control group, the mechanical property indices of each group were somewhat higher than those of the control group, yet the distinction was not substantial. This suggests that mechanical stimulation has a significant part to play in bone development.

### 2.2. MMF Increased Calcium Content and the Ability of Osteogenesis and Also Decreased the Ability of Adipogenesis in the Tibia of RL Mice

Following HE staining of the bone tissue in each mouse group, counting and quantitative statistical analysis were carried out. The results demonstrated that after 4 weeks of hindlimb unloading (HLU), the number of osteoblasts in the bone marrow was substantially decreased, while the number of adipocytes was significantly increased. After 4 weeks of recovery, in comparison with the RL group, MMF led to a remarkable increase in the number of osteoblasts in the bone tissue of RL mice (Figure 2A,C) and a reduction in the count of adipocytes in the femur bone marrow of RL mice (Figure 2B,D). Nevertheless, the number of osteoblasts did not attain the level of the age-matched mice (control group), and the number of adipocytes did not decline to the level of the age-matched mice.

By measuring the total calcium content in the tibia of each group of mice, it was found that MMF led to an increase in the total calcium content in the tibia of RL mice as compared to the RL group alone (Figure 3A). Additionally, the protein data also indicated that MMF enhanced the expression of Col1α1 protein in the femur and reduced the expression of PPARγ (Figure 3B). Through WB experiments, it was determined that MMF promoted the expression of bone formation-related proteins (OCN, OPN) and decreased the expression of the bone resorption-related proteins MMP9 and CTSK (Figure 3C,D). Since SOST acts as a negative regulator of bone formation, the WB results demonstrated that, in contrast to the RL group, MMF diminished the protein expression of SOST in the tibia of RL mice (Figure 3C,D). However, the expression levels of osteogenic and osteoclast-related proteins did not reach those of the control group. Thus, it can be concluded that MMF is capable of improving bone metabolism in the tibia of mice.

### 2.3. MMF Decreased Iron Content in the Tibia of RL Mice and Accelerated System Iron Overload Relief in HLU Mice

Upon measuring the expression levels of iron and iron-metabolism-related proteins, the results revealed that MMF reduced the total iron content in the tibia of RL mice in comparison to the RL group (Figure 4A,H), although the difference in the data was not statistically significant. The Western blot (WB) detection results of the expression levels of iron-metabolism-related proteins demonstrated that, relative to the RL group, MMF diminished the expression of the liver hepcidin protein in the duodenum (Figure 4D,E). However, it had no substantial impact on the expression of the iron-efflux-related protein FPN (Figure 4F,G). MMF also influenced the expression of iron metabolism proteins in the tibia of RL mice. Specifically, it decreased the expression of the iron-intake-related protein DMT1, increased the expression of the iron-transport-related protein TFR1, as well as the iron-storage-related protein FTH1 and the iron-efflux-related protein FPN (Figure 4B,C), which was in line with the trend of the total iron content in the tibia. This indicates that MMF plays a significant role in bone development by facilitating iron efflux, suppressing iron absorption and iron storage, thereby affecting the iron level in the bone.

### 2.4. MMF Promoted the Osteogenic Differentiation of Primary BMSCs, with Acting Wnt/β-Catenin Signaling Pathway

This section discusses the cellular-level mechanism by isolating primary BMSCs and subsequently probing into the osteoblast differentiation of BMSCs sourced from HLU mice under the influence of MMF, along with the role of iron levels and iron metabolism. The outcomes manifested that MMF stimulated the osteogenic differentiation of primary BMSCs obtained from HLU mice. Nevertheless, the osteogenic differentiation capacity did not reach that of BMSCs derived from normal mice (Ctrl group) (Figure 5A). After extracting the proteins following 14 d of osteogenic differentiation and assessing the protein expression, data analysis indicated that MMF could enhance the expression of RUNX2, OPN, and OPG during the osteogenic differentiation of HLU-derived BMSCs (Figure 5B,C). This is also correlated with the expression levels of proteins associated with the Wnt/β-catenin signaling pathway. Western blot experimental data demonstrated that MMF augmented the expression of wnt1 and P-βcatenin proteins, while MMF reduced the expression of the β-catenin and GSK-3α proteins in BMSCs derived from HLU model mice at 14 d of osteogenic differentiation (Figure 5G,H).

### 2.5. MMF Decreased the Total Iron Content in the Osteogenic Differentiation of BMSCs Derived from HLU Model Mice and Improved the Cell Iron Metabolism Level

In this section, the focus was placed on investigating the iron level and iron metabolism of BMSCs subsequent to 14 d of osteogenic differentiation induced by MMF. This was performed to unravel the alterations in iron within BMSCs and to understand the role that iron metabolism plays therein. The WB protein detection results demonstrated that MMF led to a reduction in the total iron content during the osteogenic differentiation of BMSCs (Figure 5D). The outcomes obtained from the atomic absorption method revealed that MMF decreased the expression of the iron-intake-related protein DMT1 and the iron-storage-related protein FTH1 at 14 d of osteogenic differentiation. Concurrently, MMF enhanced the expression of the iron-transport-related protein TFR1 and the iron efflux protein FPN (Figure 5E,F). These findings imply that MMF drives the differentiation of osteoblasts by curtailing the total iron content and modulating the expression of iron-metabolism-related proteins during the 14 d osteogenic differentiation of primary BMSCs derived from HLU mice.

### 2.6. MMF Inhibited the Adipogenic Differentiation of Primary BMSCs, and It Is Related to Changes in Iron Metabolism and the PPARγ Signaling Pathway

This section discusses the mechanism of the differentiation of BMSCs from HLU mice into adipocytes under MMF, and the changes in iron level and iron metabolism. The results showed that MMF inhibited the differentiation of BMSCs into adipocytes. The ORO staining results showed that MMF inhibited the formation of ORO-stained lipid droplets after 14 d of adipogenic differentiation of BMSCs derived from HLU mice (Figure 6A). Data analysis showed that MMF inhibited the expression of PPARγ in the adipogenic differentiation of HLU-derived BMSCs (Figure 6B,C) by extracting proteins after 14 d of adipogenic differentiation and detecting protein expression. In order to study the iron level and iron metabolism of BMSCs after 14 d of adipogenic differentiation by MMF, WB protein detection showed that The expression of iron-metabolism-related proteins showed that MMF inhibited the expression of the iron-intake-related protein DMT1, iron-transport-related protein TFR1, iron-storage-related protein FTH1, and iron efflux protein FPN in BMSCs after 14 d of adipogenic differentiation (Figure 6D,E). These results indicate that MMF inhibits the differentiation of adipocytes by affecting the expression of iron-metabolism-related proteins.

## 3. Discussion

Bone has the function of load-bearing and support, and its mechanical properties are important characteristics that distinguish bone from other tissues. At present, osteoporosis has become the serious disease affecting human health. Patients with osteoporosis (T ≤ −2.5) or osteopenia (−1 < T < −2.5) have reduced daily activities and are prone to fractures [30]. It is well known that the skeletal system provides mechanical support for organs, while exposure to microgravity during spaceflight leads to bone loss (T ≤ −1) [31,32]. The mice had one month-long flight experiment, and the mechanical-properties-related index of the tissue was reduced and the skeletal muscle was shrunk [33]. Therefore, it is of great significance to prevent bone loss in patients with osteoporosis and osteopenia. The composition and microstructure of bones are complex and respond to the internal and external environment. Other factors such as aging and estrogen deficiency can affect these components, ultimately leading to bone strength decreases and fracture toughness [4]. The bone matrix is composed of inorganic (i.e., mineral) and organic (i.e., water, collagen, and non-collagen) components. The role of mineral composition in bone fragility has been intensively studied and it is generally believed that, in normal bone, mineral content provides strength and stiffness. The organic matrix is thought to be responsible for the extensibility of the bone before fracture and also for absorbing energy [34]. Therefore, osteoporosis is the overall manifestation of a series of macro and micro events, and the mechanical property of the bone is the ultimate manifestation of the load-bearing effect, so the study of it has a more important research value. Therefore, in this paper, our experiment mainly focuses on the mechanism of bone mechanics in mice. The core focus of the paper is to explore the mechanical properties of osteoporosis mice following a physical source, and the index data are developing in a favorable direction. The basic research and clinical use of osteoporosis have achieved a lot, but there is still a lack of specific drugs for prevention and treatment, so the research on osteoporosis has always been an important task for scientists. The population of the HLU reloading study in this paper includes not only menopausal women and senile osteoporosis patients, but also astronauts who have engaged in space missions.

The thermal effect of the electromagnetic field, combined with the complex environment, renders the discussion of mechanisms highly challenging. Consequently, in recent years, numerous studies on the biological effects of a static magnetic field, as a solitary physical source, on tissues and cells have emerged successively. Research on the treatment of bone-remodeling-related diseases, such as osteoporosis, using a static magnetic field has mainly centered around fracture and bone healing. Presently, an increasing amount of research is being dedicated to the study of a static magnetic field (SMF) for osteoporosis. Our review has summarized the application of SMF in clinical osteoporosis patients [35]. The hypomagnetic field leads to an augmentation in gene expression within MC3T3-E1 cells, subsequently inhibiting the differentiation of osteoblasts. The hypomagnetic field (HyMF) can also promote the accumulation of iron in the liver and bone, thereby disrupting the amelioration of iron accumulation [36]. The impact of SMF on different cells varies. Among them, the moderate magnetic field has attracted extensive attention from scientific researchers due to its pronounced effects and the relative ease of clinical translation and equipment realization. Studies have demonstrated that SMFs (ranging from 0.2 T to 0.6 T) inhibit the adipogenic differentiation of bone marrow stromal cells (BMSCs), while promoting their osteogenic differentiation in a manner dependent on magnetic field strength. Genome-wide RNA sequencing and bioinformatics analysis have revealed that SMF (at 0.6 T) reduces PPARγ-mediated gene expression but enhances Runx2-mediated gene transcription in BMSCs. Additionally, SMFs significantly mitigate bone mass loss induced by dexamethasone or all-trans retinoic acid in mice [28].

There has been a lot of progress in the study of the HLU mice experiment. Yang et al. summarized the research progress of SMF on bone diseases and bone-tissue-related cells in recent years, and also proved that different SMFs have different effects on different model mice and bone-tissue-related cells [37]. Zhang et al. proved that 1 T–2 T SMF can prevent osteoporosis in OVX mice [38]. Although there has been much research progress and many achievements, there are challenges in the use of SMF is due to its multiple targets and the complexity of osteoporosis disease. Recently, the effects have not been entirely consistent and the mechanisms involved have also not been fully elucidated, while due to the different magnetic field intensity, gradient, direction, exposed time, and prevention and treatment effects, the mechanisms of SMF on HLU reloading mice are not completely the same. In this study, MMF was used to investigate the effects on the bone quality of HLU reloading mice. The results showed that the mechanical properties were recovered compared to the control group, and the mechanical properties recovered better than in GMF recovery [39]. Zhang et al. proved that 4 mT SMF exposure for 16 weeks inhibited the architectural deterioration of the trabecular bone and cortical bone, and Lv et al. showed similar results in a study of the metabolic condition in T1DM rats [40]. In this study, the calcium content of the tibia was also increased by MMF treatment in contrast to the GMF group. In addition, our results demonstrated that MMF also promoted the expression of bone-formation-related proteins in the tibia and the expression of P1NP in the serum. The expression of bone-resorption-related protein was inhibited, and the expression of Sost protein was decreased. Meanwhile, the expression of CTX-I in the serum was also decreased. Lv et al. has shown that the metabolic condition in type 1 diabetes mice inhibited bone absorption, and Zhang et al. also proved that SMF could promote bone formation [41]. Bone tissue was stained by immunohistochemistry; the results showed that the expression of Col1αI proteins was increased, and the expression of PPARγ was decreased. These data indicated that MMF promoted bone metabolism and improved the mechanical properties of the tibia.

Iron is of great significance in the normal development of the body and is involved in numerous essential physiological processes, including oxygen transport, DNA synthesis, and mitochondrial electron transport. It is also a crucial element in collagen production. Malfunctions in certain key aspects can severely disrupt the balance of iron metabolism and give rise to a variety of diseases associated with iron metabolism imbalance. Iron overload is highly correlated with osteoporosis in postmenopausal women and astronauts [42]. In a 3-year study involving 940 healthy postmenopausal women, a decrease in femoral bone mineral density (BMD) was linked to an increase in serum ferritin, which can accelerate bone loss and heighten the risk of hip fracture [43]. Xu et al. have demonstrated that iron accumulation is closely related to the occurrence and progression of osteoporosis and is an independent risk factor for it [44]. Our data indicated that the iron content in the tibia of hindlimb unloading (HLU) mice was higher than that of the control group. MMF was capable of reducing the iron content in the tibia of HLU reloading mice and enhancing the bone metabolism of HLU mice. We also discovered that MMF could modulate the expression of iron-metabolism-related proteins. This research concurred with the finding reported by Zhang et al. that hepcidin, as a key regulator of iron homeostasis, protected against osteoporosis by decreasing the iron content in bone tissue [45]. In this paper, the regulation of iron metabolism can improve the restoration of mechanical properties in HLU reloading mice. Specifically, MMF promoted the expression of ferroportin (FPN) in the duodenum of HLU reloading mice, yet it had no significant impact on the hepcidin level in the liver.

Similarly, cellular iron metabolism is of fundamental importance for the growth and development of an organism. MMF modulated the expression of iron-metabolism-related proteins in primary BMSCs following 14 days of differentiation. It was able to enhance bone formation, suppress the differentiation of adipocytes, and rectify the imbalance in bone homeostasis after HLU. MMF diminished the total cellular iron content by repressing DMT1 expression and augmenting FPN expression, which is largely in line with the normal iron metabolism process. Consequently, it can be asserted that the osteogenic differentiation potential of BMSCs can be influenced by regulating the expression of iron-metabolism-related proteins. Dong et al. [43] investigated that a 16 T SMF reduced the protein expression of DMT1, FTH1, and FPN. However, the underlying mechanism differs from ours. Specifically, our study indicates that MMF decreased the protein expression of DMT1 and FTH1, while increasing the protein expression of TFR1 and FPN. The disparity arises due to the distinct SMF parameters. Additionally, it has been reported that HyMF exacerbates the symptoms of osteoporosis by impacting the expression of systemic iron metabolism [45].

The Wnt/β-catenin signaling pathway plays an important role in bone growth and development. It is one of the Wnt signal transduction pathways which leads to the accumulation of β-catenin in the cytoplasm and translocation to the nucleus, thus initiating the osteogenic differentiation of downstream BMSCs. The Wnt/β-catenin signaling pathway can adjust the direction of BMSC differentiation. The mechanism is that activating the Wnt/β-catenin signaling pathway can promote osteogenic differentiation by regulating the intrinsic potential and direction of the pre-differentiation of BMSCs. In this paper, the results showed that MMF activated the Wnt/β-catenin signaling pathway to a certain extent and promoted the osteogenic differentiation of BMSCs. Shen et al.’s studies showed that low Foxf1 could promote BMSCs towards osteogenetic differentiation, activate the Wnt/β-catenin signaling pathway, and prevent the bone loss induced by OVX [46]. The expression of PPARγ is full in adipocytes and it plays a key role in the adipogenic differentiation of BMSCs. It is one of the potential effective ways to treat osteoporosis by regulating the PPARγ signaling pathway and bone metabolism. This paper showed that MMF inhibited the adipogenic differentiation of BMSCs derived from HLU model mice, not only in ORO staining, but also in the expression of the PPARγ protein. Therefore, MMF inhibited the adipogenic differentiation of BMSCs derived from HLU, and then restarted the progression of osteoporosis, Chen et al.’s studies have shown that 0.2 T–0.6 T SMF inhibited BMSCs into adipocytes and also the expression of related transcription factors [28]. Thus, this paper can understand the mechanism of the occurrence and development of osteoporosis at the molecular level and provide an important theoretical basis for the treatment of osteoporosis (Figure 7).

## 4. Materials and Methods

### 4.1. Animals and Treatments

Adult seven-week-old male C57BL/6 mice (Charles River Laboratories, Beijing, China) were used in this study. The study was approved by the Animal Ethics Committee of Northwestern Polytechnical University, and carried out in compliance with the ARRIVE guidelines 2.0 (https://arriveguidelines.org, accessed on 14 July 2020). All mice were housed at 24 ± 2 °C, maintained under 12 h light/dark cycle conditions, and allowed free access to tap water and chow. All mice were acclimated for 7 days before starting the experiment. Mice were subjected to hindlimb unloading (HLU) for 4 weeks followed by 4 weeks of reloading in GMF and MMF (Figure 1A,B). Mice were randomly divided into five equal groups as follows: the control 1 group (Ctrl 1, the mice exposed to GMF for 4 weeks), the hindlimb unloading 1 group (UL1, HLU, the mice were unloaded for 4 weeks), the control group (Ctrl, the mice exposed to GMF for 8 weeks), the reloading group (RL, the mice were unloaded for 4 weeks and then reloaded in GMF for another 4 weeks), and the reloading + MMF group (RL + MMF, the mice were unloaded for 4 weeks and then reloaded in MMF for another 4 weeks). At the end of the experiment, mice were anesthetized and subjected to collect blood samples via cardiac puncture. Left femurs were fixed in 4% paraformaldehyde for 2 days, and then transferred to phosphate-buffered saline for histochemical analysis. Right femurs were wrapped in saline-soaked gauze and stored at 80 ℃ until used for mechanical properties testing.

### 4.2. Reagents

The CD105, CD90, CD73, CD34, and CD45 antibodies were purchased from BioLegend (San Diego, CA, USA). Vitamin C and sodium β-glycerophosphate were purchased from Sigma-Aldrich (Merck KGaA, Darmstadt, Germany); dexamethasone and the ALP activity detection kit were purchased from Beyotime Biotechnology (Shanghai, China). The mouse PINP Elisa kit, Mouse CTX-I Elisa kit, and Mouse Ferrritin Elisa kit were purchased from Nanjing Jiancheng Technology Co., Ltd. (Nanjing, China). DMT1, FPN1, Col1α1, OPN, MMP9, CTSK, PPAR-γ, β-catenin, and Wnt1 were purchased from Proteintech Group, Lnc. (Chicago, IL, USA); FTH1 and Sost were purchased from Abcam (Cambridge, UK); Steap3, TFR1, hepcidin, OCN, and LRP6 were purchased from Abclonal (Wuhan, China); OPG was purchased from Affinity; RUNX2, P-β-catenin, and GAPDH were purchased from Cell Signaling Technology (Boston, MA, USA); Gsk-3α was purchased from Signalway Antibody (Greenbelt, MD, USA). All reagents used for Western blotting and immunostaining were configured in strict accordance with the instructions and stored at 4 °C for later use.

### 4.3. Static Magnetic Field Exposure Systems

The 0.2 T–0.4 T SMF was generated in a traffic circle around a superconducting magnet (JASTEC, Kobe, Japan) with the highest magnetic field intensity of 16 T at the center. The magnetic field strength weakened with distance away from the center of magnet. The mice were housed in cages made of polymethyl methacrylate. The cages were a 1/4 arc shape with an inner diameter of 400 mm, an outer diameter of 500 mm, and a height of 160 mm. In the annular space from the center 400 mm to 500 mm, the distribution of the magnetic field was about 0.2 T–0.4 T with a decreased gradient of 2.09 T/m along the radial site from 400 mm to 500 mm at the radial distance from the center of the superconducting magnet [47]. All cages were placed on a shelf at the same height, and to match their biological rhythms, MMF treatment was conducted from 8 am to 8 pm.

### 4.4. Mechanical Properties

The mechanical properties of the femur were detected using a three-point bending test by the Instron-5943 Universal Material Testing Machine (Instron 5943; Instron, Canton, MA, USA). The tibia was secured on the two supports of the anvil on the machine. The span of two supports was 8 mm and the loading rate was 1 mm/min until the tibia fracture. The stress–strain curve was obtained, and the data of the inner diameter and outer diameter of the cross section of the fractured tibia were obtained by using the ultra-depth of field microscope. Finally, the performance parameters of the material were obtained by using the MATLAB R2024b software (MathWorks Inc., Natick, MA, USA) to calculate and normalize the data. These include the stiffness, ultimate load, ultimate stress, elastic modulus, yield stress, and toughness overall.

### 4.5. Measurement of Iron/Calcium Elements

The tibias were dried overnight and ash treatment was carried out in a resistance furnace. Nitrification was performed by using HN0_3_ at 70 °C. Subsequently, a mixture of 1% HNO_3_ and 0.1% KCl was used to dilute to 10 mL. The atomic absorption spectrometer flame method was used to detect the levels of iron/calcium by atomic absorption spectroscopy (AAS; Analytik Jena AG, Jena, Germany). The iron/calcium content was statistically analyzed by the ratio of iron/calcium content to dry weight. Also, BMSCs were treated with osteogenesis/adipogenic induction medium for 14 d and cells were divided into two equal parts for protein quantification and iron quantification. The cell samples for protein quantification were lysed by RIPA and analyzed with a BCA protein assay kit. The total iron level was determined by AAS and normalized to the protein concentration.

### 4.6. Western Blotting Analysis

The bone marrow of mouse tibia tissue was washed out and discarded at low temperature, added with RIPA and lysed at low temperature, then centrifuged at 12,000× *g* for 10 min at 4 °C. The primary BMSCs were extracted and purified by Ficoll solution and an adherent method. After the cells were passaged from passage 2 to passage 4, BMSCs were treated with induction medium for 14 d. Cells were harvested and protein concentration was measured by a BCA kit assay, before 25 μg protein was loaded onto 10% SDS-PAGE membranes. According to the molecular weight of the target protein, the polyvinylidene fluoride (PVDF) membrane was attached to the corresponding molecular weight for membrane transfer, blocked in 5% skim milk, and incubated with primary antibodies against hepcidin, FTH1, GAPDH, Col1α1, OCN, OPN, SOST, MMP9, CTSK, DMT1, FTH1, TFR1, FPN, OPG, RUNX2, LRP6, β-catenin, P-β-catenin, Wnt1, GSK-3α, and PPARγ at 4 °C. After washing with TBST, the membrane was incubated with horseradish peroxidase (HRP)-conjugated secondary antibody (Beyotime, Shanghai, China) for 2 h at room temperature; the image was captured by chemiluminescence system (T5200Multi; Shanghai Peiqing Technology Co., Ltd., Shanghai, China) and analyzed quantitatively for protein bands by Image J1 (National Institutes of Health, Bethesda, MD, USA; https://imagej.net/ij/).

### 4.7. Assessment Osteogenic/Adipogenic Differentiation of Primary BMSC

Primary BMSCs were grown in basal medium until appropriate cell confluence (approximately 80–90%). For osteogenic differentiation, BMSCs were cultured in osteogenic induction medium containing 50 mM ascorbate-2-phosphate, 0.1 mM dexamethasone, and 10 mM β-glycerol phosphate for 14 d. The culture medium was changed every three days. After treatment, BMSCs were fixed with 4% PFA before ALP staining. At the end of differentiation, alizarin red S (ARS) staining was performed to evaluate the mineralization of the cell matrix. For adipogenic differentiation, BMSCs were cultured in adipogenic induction medium for 14 d. The culture medium was changed every third day. After treatment, BMSCs were fixed with 4% PFA before ORO staining. At the end of differentiation, ORO staining was performed to evaluate the adipogenic differentiation ability of the cell.

### 4.8. Histological Analysis

The mice were euthanized and the left tibias of the mice were obtained and fixed in 4% PFA for 48 h. The tibia was decalcified by 10% EDTA with a replacement of fresh EDTA solution every three days continuously for thirty days. Following decalcification, the tibia was embedded in paraffin and processed for paraffin sections with 5 μm via a semiautomated rotary microtome. Then, the samples were stained with hematoxylin and eosin (H&E) for morphological analysis and to assess the number or area of osteoblasts and adipocytes; Prussian blue staining was used for the analysis of iron-metabolism indexes. For immunohistochemistry staining, deparaffinized and dehydrated tissue sections were re-hydrated before being subjected to antigen retrieval, and then blocked with diluted normal serum for at least 1–1.5 h at 25 °C to eliminate nonspecific binding. The slides were incubated with primary antibodies against Col1α1, OCN, SOST, and PPARγ overnight at 4 °C. After careful washing, the sections were incubated with horseradish peroxidase conjugates to detect positive signals. Slides incubated with polyclonal rabbit IgG (Abcam) served as negative controls. Pictures were captured and monitored using a Leica Microsystems microscope (Leica Microsystems Ltd., Wetzlar, Hesse, Germany).

### 4.9. Serum Biochemical Assay

The blood samples were centrifuged at 3000 rpm for 15 min at 4 °C, and the serum was fractionated. Serum markers for bone turnover, including serum P1NP, CTX-I, and ferritin were examined by using respective mouse enzyme-linked immunosorbent assay (ELISA) kits (Nanjing Jiancheng Technology Co., Ltd., Nanjing, China).

### 4.10. Statistical Analysis

Statistical analysis was performed by using the GraphPad Prism 6.0 (GraphPad Software, Inc., La Jolla, CA, USA) and the results are presented as the mean ± standard error of the mean. The significance of differences between different experimental groups was determined by using one-way ANOVA with Fisher’s least significant difference (LSD) multiple comparisons test. Values of *p* < 0.05 were considered statistically significant.

## 5. Conclusions

The use of 0.2 T–0.4 T SMF improved the bone quality of the tibia in HLU reloading mice, especially in mechanical properties, which was closely related to the regulation of iron metabolism. Moreover, through the mechanistic study of in vitro experiments, 0.2 T−0.4 T SMF had an essential role in the improvement of the mechanical properties of mice by regulating the level of iron metabolism in the osteogenic and adipogenic differentiation of primary BMSCs from HLU model mice. It not only meant that 0.2 T−0.4 T SMF could be used for the treatment of osteoporosis but also in the auxiliary equipment for astronauts returning to the ground to provide an experimental data basis. Moreover, it created new ideas for the treatment of osteoporosis in astronauts and postmenopausal and elderly people on the ground, and the study of the mechanism of iron metabolism has also opened new thoughts for treatment in the future.

## Figures and Tables

**Figure 1 ijms-25-13136-f001:**
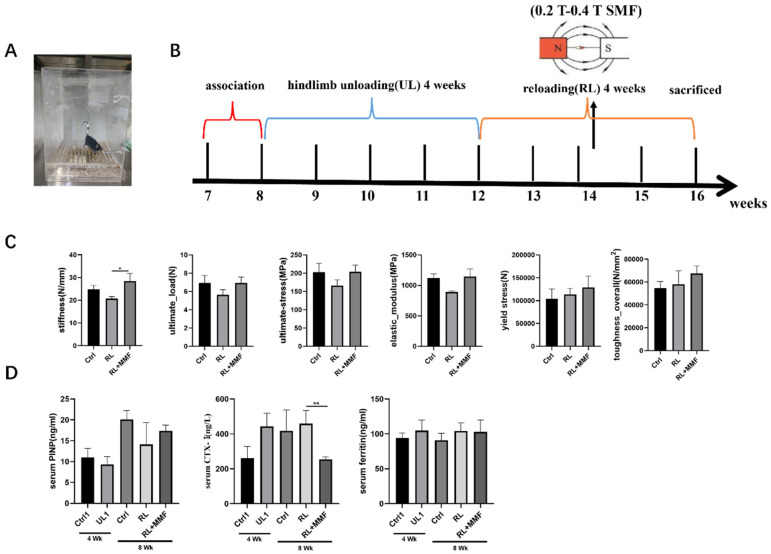
Effects of MMF on mechanical properties and markers of bone metabolism in mice. (**A**) mouse tail suspension model, (**B**) experimental flow chart, (**C**) mechanical properties of mouse tibia, including stiffness, elastic modulus, ultimate strain, ultimate stress, ultimate load, and ultimate displacement, (**D**) measurement of serum biochemical indexes. *n* = 4–6. Data are shown as the mean ± SEM. * *p* < 0.05, ** *p* < 0.01.

**Figure 2 ijms-25-13136-f002:**
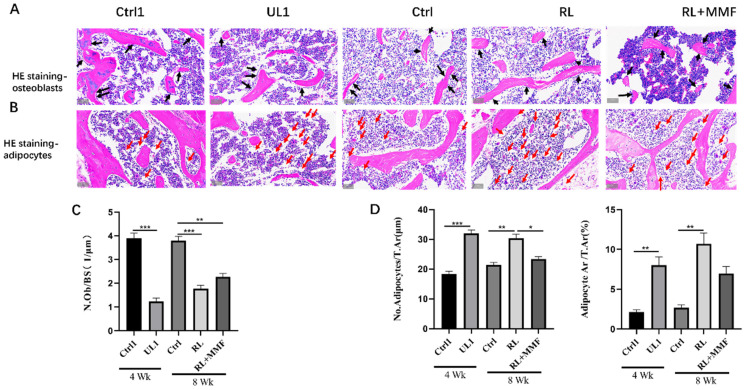
The number of osteoblasts and adipocytes in the bone marrow of femur were evaluated. (**A**,**B**) The femoral sections were subjected to H&E staining to visualize osteoblasts and adipocytes. Black arrows indicate osteoblasts. Red arrows indicate adipocytes. Scale bar = 50 μm. Osteoblastogenesis were evaluated by the osteoblast number per bone surface (N.Ob/BS) in the trabecular bone; (**C**) Adipogenesis was evaluated by the adipocyte number per bone surface (No. Adipocytes/T.Ar) and adipocyte surface per bone surface (Adipocyte.Ar/T.Ar) in the bone marrow (**D**). *n* = 3. All the data are shown as mean ± SEM. * *p* < 0.05, ** *p* < 0.01, *** *p* < 0.001.

**Figure 3 ijms-25-13136-f003:**
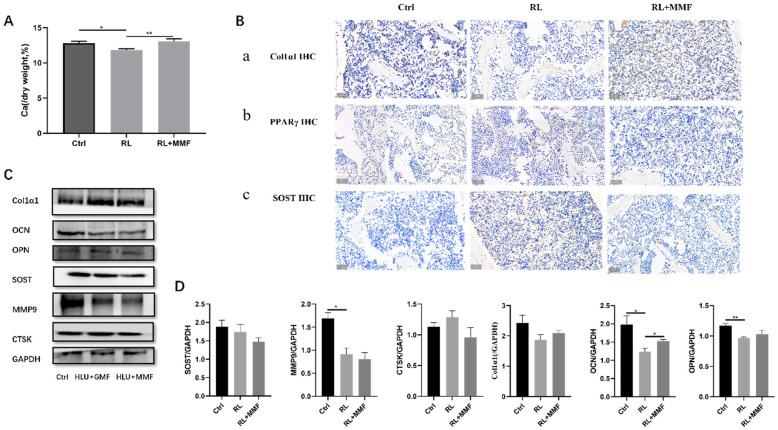
Bone metabolism of MMF on the femur of HLU reloading mice. (**A**) Total calcium content in the tibia of HLU reloading mice. (**B**) Immunohistochemical staining of Col1α1, PPARγ, and SOST. Scar bar = 50 μm. Brown color indicates positive expression. (**C**) Protein expression of osteogenic and osteoclast-related proteins in the tibia of HLU reloading mice; (**D**) Quantitative statistics of protein expression of (**A**). *n* = 4–6. Data are shown as the mean ± SEM. * *p* < 0.05, ** *p* < 0.01.

**Figure 4 ijms-25-13136-f004:**
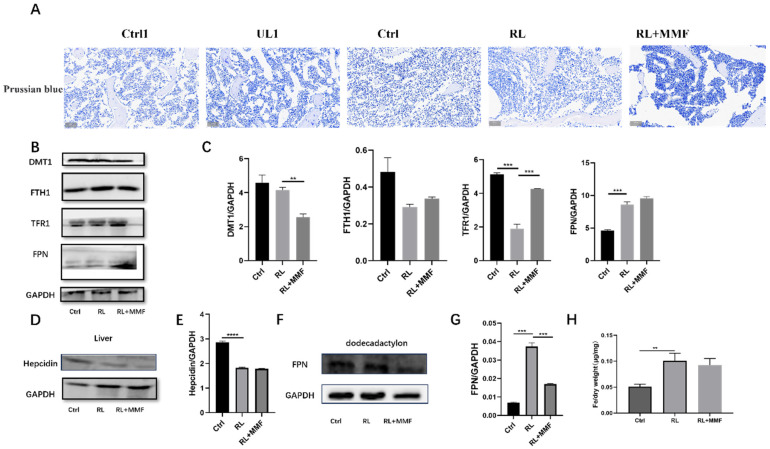
Effects of MMF on iron metabolism in HLU reloading mice. (**A**) Prussian blue staining of bone tissue in HLU reloading mice, Scar bar = 50 μm; (**B**) Protein expression related to iron metabolism in the tibia of HLU reloading mice; (**C**) Quantitative statistics of the protein expression of (**B**); (**D**) Protein expression of liver tissue in HLU reloading mice; (**E**) Quantitative statistics of the protein expression of (**D**); (**F**) Protein expression of the duodenum in HLU reloading mice; (**G**) Quantitative statistics of the protein expression of (**F**); (**H**) Total iron content in the tibia of HLU reloading mice. *n* = 3. Data are shown as the mean ± SEM. ** *p* < 0.01, *** *p* < 0.001, **** *p* < 0.0001.

**Figure 5 ijms-25-13136-f005:**
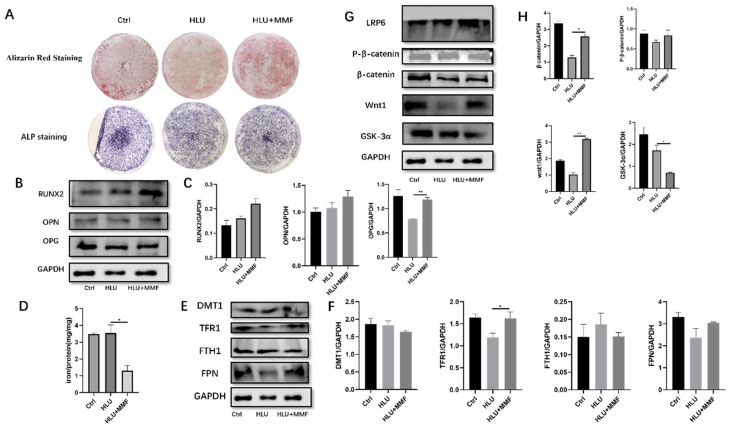
MMF promotes osteogenic differentiation of primary BMSCs and mechanism. (**A**) ARS and ALP staining after 14 d of osteogenic differentiation of BMSCs, 10×; (**B**) Protein expression of osteogenic related molecules after 14 d of osteogenic differentiation of BMSCs; (**C**) Quantitative statistics of the protein expression of (**B**); (**D**) Total iron content of BMSCs cells after osteogenic differentiation for 14 d; (**E**) Expression of iron-metabolism-related proteins of BMSCs after osteogenic differentiation for 14 d; (**F**) Quantitative statistics of the protein expression of (**E**); (**G**) Protein expression of the Wnt/β-catenin signaling pathway after 14 d of osteogenic differentiation of BMSCs; (**H**) Quantitative statistics of the protein expression of (**G**). *n* = 3. Data are shown as the mean ± SEM. * *p* < 0.05, ** *p* < 0.01.

**Figure 6 ijms-25-13136-f006:**
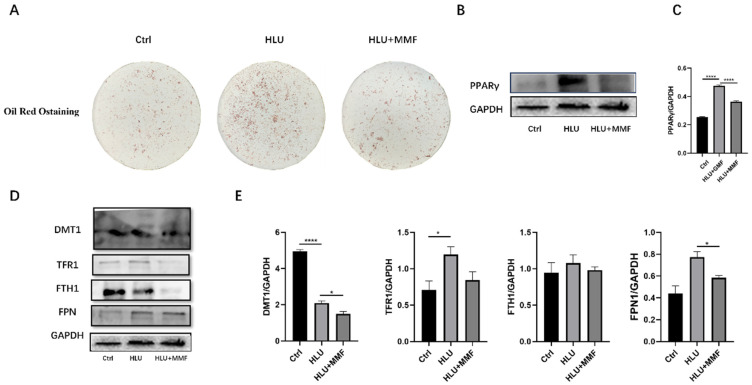
MMF inhibits adipogenic differentiation of primary BMSCs derived from HLU model mice. (**A**) Oil Red O staining after 14 d of adipogenic differentiation of BMSCs, 10×; (**B**) Protein expression of PPARγ after 14 d of adipogenic differentiation of BMSCs; (**C**) Quantitative statistics of the protein expression of (**B**); (**D**) Expression of iron-metabolism-related proteins of BMSCs after adipogenic differentiation for 14 d; (**E**) Quantitative statistics of the protein expression of (**D**). *n* = 3. Data are shown as the mean ± SEM. * *p* < 0.05, **** *p* < 0.0001.

**Figure 7 ijms-25-13136-f007:**
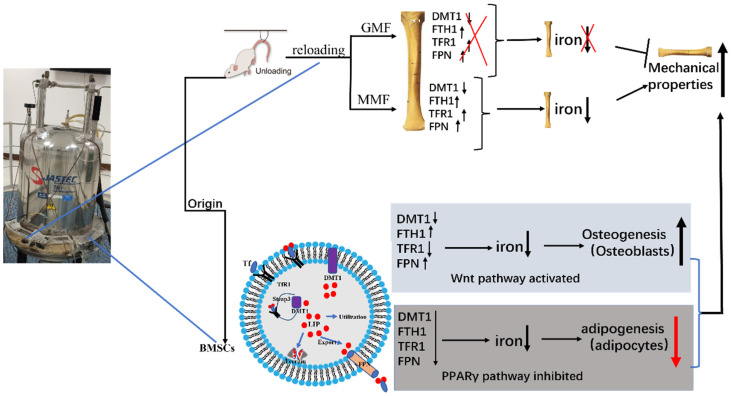
Schematic diagram of results summarized in this paper. The use of 0.2 T–0.4 T SMF (MMF) improved the bone quality of HLU reloading mice by affecting bone metabolism and iron metabolism. Through study of the osteogenic and adipogenic differentiation of primary BMSCs, it was proved that MMF promoted osteogenic differentiation and activated the Wnt/β-catenin signaling pathway, inhibiting adipogenic differentiation. Upward arrows represent an increase; downward arrows represent a decrease. All the colors have no real meaning, just for beauty.

## Data Availability

Data are provided within the manuscript.

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
