# Peer review of "A 0.2 T–0.4 T Static Magnetic Field Improves the Bone Quality of Mice Subjected to Hindlimb Unloading and Reloading Through the Dual Regulation of BMSCs via Iron Metabolism"

_ijms, 2024, doi:10.3390/ijms252313136_

Round 1

Reviewer 1 Report

Comments and Suggestions for Authors

This paper addresses the effect of unloading and loading in of mice hindlimbs in the iron contents of the tissue by applying a static magnetic field. The work here has many aspects and it was examined both from the mechanical and biological point of view. However, this is not a very well presented work. In fact, it is not written in a very appropriate way. The impact of the work is not made clear enough.

Here are some comments:

Section 2.2: this needs to be rewritten, this is not how methods are presented.

Section 2.4: This is a better written section. However, mechanical properties are not detected, they are, investigated/assessed/obtained. It says that the mechanical test was conducted until the tibia fractured. Is this completed failure or cracking? How was the machine controlled? Bone is an anisotropic heterogeneous material, when you are talking about the mechanical properties, the ones that were obtained here are at a macro structural level. This is particularly important when referring to stiffness, modulus and toughness, which are properties defined at the nanolevel of the material.

Section 3.1, 3.2 section titles are not supposed to be whole sentences

The graphs in the results are very hard to read and not very well explained in the text. The mechanical properties for exampled aren’t stated in the text.

The discussion needs to be more critical of the experimental work done and the results.

Figure 7 could be part of the methods or results but definitely not the conclusions. 

Comments on the Quality of English Language

Line 39: rewrite this sentence there’s no flow to the text.

Line 70: this is the first time, other than the abstract that you are mentioning SMF. You need to write the whole words again.

Line 92: The aim is not clear or well written.

Line 100: This is not how websites should be referenced.

Line 103: write the whole words for HLU
Line 113: -80 not 80

Line 118: Companies’ names need to at least be written with a capital letter at the beginning of the word. Additionally for each company you need to mention the country of origin in brackets next to the name

Line 188: the methods need to be written in 3rd person.

Reviewer 2 Report

Comments and Suggestions for Authors

The article raises an interesting issue. The research was conducted in a very broad manner and its plan and execution should be assessed highly. The discussion was conducted properly and the summary is clear.

The Authors should be congratulated on the execution of a comprehensive research plan and its proper description.

My comments concern the presentation of the results. It seems appropriate to include illustrations (diagrams) and photos from the performance of the strength tests themselves, but also others. It is interesting how the bones were placed in the strength machine in a repeatable manner.

The presentation of the data is difficult to observe and verify because the diagrams are very small and therefore not very legible even when enlarged. This should be corrected.

This does not change my high assessment. After making corrections and additions, the article can be published.
